# Personalised Therapies for Metastatic Triple-Negative Breast Cancer: When Target Is Not Everything

**DOI:** 10.3390/cancers14153729

**Published:** 2022-07-31

**Authors:** Serena Capici, Luca Carlofrancesco Ammoni, Nicole Meli, Viola Cogliati, Francesca Fulvia Pepe, Francesca Piazza, Marina Elena Cazzaniga

**Affiliations:** 1Phase 1 Research Centre, ASST-Monza (MB), 20900 Monza, Italy; serena.capici@gmail.com (S.C.); viola.cogliati.vc@gmail.com (V.C.); francesca.pepe@asst-monza.it (F.F.P.); marina.cazzaniga@unimib.it (M.E.C.); 2School of Medicine and Surgery, University of Milano-Bicocca, 20900 Monza, Italy; nicolemeli1994@gmail.com (N.M.); f.piazza005@unibs.it (F.P.)

**Keywords:** TNBC, androgen receptor, PI3K/AKT/mTOR, metronomic

## Abstract

**Simple Summary:**

The purpose of the present review is to shed light on new molecular biomarkers in triple-negative breast cancer (TNBC), showing emerging therapeutic approaches related to specific molecular signatures and their mechanisms of action. A general overview of ongoing clinical trials, future perspectives and differences in approval by American and European regulatory authorities is provided.

**Abstract:**

Triple-negative breast cancer—defined by the absence of oestrogen/progesterone receptors and human epidermal growth factor receptor 2 expression—is a complex and heterogeneous type of tumour characterised by poor prognosis, aggressive behaviour and lack of effective therapeutic strategies. The identification of new biomarkers and molecular signatures is leading to development of new therapeutic strategies including immunotherapy, targeted therapy and antibody-drug conjugates (ADCs). Against a background where chemotherapy has always been considered the standard of care, evolution towards a precision medicine approach could improve TNBC clinical practice in a complex scenario, with many therapeutic options and new drugs. The aim of this review was to focus on emerging therapeutic targets and their related specific therapy, discussing available and emerging drugs, underlining differences in approval by American and European regulatory authorities and showing the future perspective in the large number of ongoing clinical trials.

## 1. Introduction and Background

Breast cancer (BC) is the most common malignant neoplasm in women. About 2.3 million new BC diagnoses and 685,000 BC-related deaths occurred worldwide in 2020 [1]. Triple-negative breast cancer (TNBC)—defined by the absence of the oestrogen receptor/progesterone receptor and human epidermal growth factor receptor 2 (HER2) expression—represents approximately 15% of invasive BC. It is predominantly associated with pre-menopausal status, African–American ethnicity and women with BRCA 1–2 germline mutations. It is characterised by a more aggressive clinical course, including an advanced stage at initial diagnosis, and earlier recurrence with metastatic spread, and it has a poor prognosis with median overall survival (OS) rarely extending beyond 12 to 18 months in advanced disease [2,3]. In patients with locally recurrent inoperable or metastatic disease (mTNBC), treatment options have been represented by chemotherapy, in particular anthracyclines, taxanes, capecitabine eribulin and platinum used as monotherapy or in combination for a very long time. Unfortunately, despite the use of these drugs, the prognosis of TNBC remains poor, and new approaches able to overcome this issue are necessary. An answer to this need may be found in precision medicine, a medical model that proposes therapies tailored to patients’ characteristics. Indeed, although TNBC has been historically considered a unique entity defined by immune-histochemistry features, from a molecular point of view it is a heterogeneous entity characterised by different gene expression patterns. Lehmann, et al., identified different molecular subtypes of TNBC, which include basal-like 1 (BL1), basal-like 2 (BL2), immunomodulatory (IM), mesenchymal (M), mesenchymal stem-like (MSL), and luminal androgen receptor (LAR). The BL1 and BL2 subtypes present higher expression of DNA damage response genes; the M and MSL subtypes are enriched in gene expression for epithelial–mesenchymal transition; the LAR subtype is characterised by androgen receptor (AR) signalling [4]. Thus, based on gene expression, about 30% of TNBCs do not overlap with the basal-like intrinsic subtype, and this heterogeneity can explain why patients with MTNBC have different clinical outcomes. Furthermore, each subtype is characterised by a distinct gene expression signature that can justify a tailored approach leading to the selection of specific therapies. In the past few years, many efforts have been made to identify molecular signatures and targets that could improve TNBC clinical practice, resulting in a messy and ever-changing landscape of therapeutic options. Moreover, the differences in approval by American and European regulatory authorities (the Food and Drug Administration and the European Medicines Agency) favour confusion regarding the possibility of access to drugs in different countries.

In this review, we aimed to put order into this complex scenario by focusing on emerging therapeutic targets and their related specific therapies. Notably, we articulated the discussion according to the different approaches that could be used to interfere with tumour mechanisms, taking into account both cellular and microenvironmental pathways. For each approach, we focused on specific therapeutic targets, and discussed available and emerging drugs. Figure 1 schematically represents where the different targets are located, to help the reader’s comprehension.

## 2. Targeting Microenvironment (Angiogenesis and Immune System)

### 2.1. VEGFR

Vascular endothelial growth factor (VEGF) is expressed in 30–60% of TNBC [5]. VEGF promotes angiogenesis by stimulating endothelial cell proliferation and migration, inhibiting endothelial cell apoptosis, and supporting the newly formed blood vessels. As angiogenesis is considered a key component for tumour cell proliferation and survival, bevacizumab, a monoclonal antibody that binds to VEGF-A, has been studied in several phase III trials in addition to first-line or second-line chemotherapy treatment in metastatic breast cancer (MBC). 

In a phase III trial comparing bevacizumab + paclitaxel vs. paclitaxel alone in first-line treatment for HER2-negative MBC (E2100 trial), the addition of bevacizumab improved Progression-Free-Survival (PFS), but no difference was observed in OS. The result was confirmed in the subgroup analysis, as the TNBC patients (about one third of enrolled patients) achieved PFS of 8.8 vs. 4.6 months (HR 0.53), favouring the combination of paclitaxel and bevacizumab [6]. 

Similarly, the randomised, double-blind, phase III study AVADO, which investigated the combination of bevacizumab (7.5 mg/kg or 15 mg/kg) with docetaxel in women with HER2-negative MBC who had not received chemotherapy for metastatic disease, showed superior PFS with the combination, but similar results in terms of OS; the TNBC subpopulation analysis for PFS was consistent with the results for the overall study population [7].

The RIBBON-1 and RIBBON-2 trials evaluated the efficacy and safety of bevacizumab in combination with investigator’s choice of chemotherapy vs. chemotherapy-plus-placebo, respectively, as first-line and second-line treatment of HER2-negative MBC. Both trials showed improved PFS and Overall Response Rate (ORR) with the combination treatment, but no differences were observed in terms of OS. The PFS benefit from bevacizumab was observed consistently in all pre-specified and exploratory subgroups, including triple-negative disease [8,9].

The TANIA trial—which evaluated bevacizumab + chemotherapy vs. chemotherapy alone in second-line treatment cancer after first-line therapy with bevacizumab plus chemotherapy in an HER2-negative population—demonstrated that in patients with bevacizumab-pre-treated MBC the continuation of bevacizumab significantly improved second-line PFS; however, further bevacizumab after the second progression did not significantly improve third-line PFS or OS [10].

The results of the GINECO A-TaXel phase II study demonstrated that a triplet regimen of paclitaxel, capecitabine and bevacizumab, followed by maintenance therapy with capecitabine and bevacizumab, had a high activity and a manageable safety profile in the mTNBC population [11].

In the randomised, non-inferiority, phase III trial (TURANDOT trial), bevacizumab + capecitabine represented a valid first-line treatment option for HER2-negative MBC, offering good tolerability without compromising OS when compared with bevacizumab + paclitaxel, although in the TNBC subpopulation, median OS was lower in the arm of patients who received bevacizumab + capecitabine (17.7 vs. 24.4 months) [12].

Based on the results of the E2100 trial, bevacizumab in combination with paclitaxel was approved by the FDA and the EMA as first-line treatment for patients with MBC. Moreover, bevacizumab in combination with capecitabine was approved as first-line treatment of MBC patients for whom therapy with other chemotherapy regimens, including those based on taxanes or anthracyclines, was not considered appropriate. Nevertheless, in November 2011, the FDA announced that breast cancer indication for bevacizumab had been withdrawn, after concluding that the improvement in PFS observed in the AVADO and RIBBON 1 studies were not as consistent as the PFS improvement observed in the E2100 trial; furthermore, the addition of bevacizumab to chemotherapy resulted in an increased rate of serious adverse events (grade 3–5), some of which directly attributable to bevacizumab in both studies [13].

Further studies about the role of VEGFR inhibitors in combination with other agents for mTNBC are ongoing, such as the ATRACTIB phase II trial evaluating the efficacy and safety of first-line atezolizumab in combination with paclitaxel and bevacizumab (ClinicalTrials.gov Identifier: NCT04408118), and the NCT05192798 phase II trial investigating the first-line combination of bevacizumab and nab-paclitaxel (ClinicalTrials.gov Identifier: NCT05192798). 

Table 1 summarises the most relevant results and the most important ongoing trials, regarding bevacizumab.

### 2.2. Targeting Immune System

TNBC has an immunogenic microenvironment that contains a variety of cell types, including tumour-infiltrating lymphocytes (TILs), tumour-associated macrophages (TAM), neutrophils and fibroblasts, and NK cells [23]. It has been found that tumours with high levels of TILs—including CD20+ B cells, CD4 and CD8+ T cells, and CD138+ plasma cells—have better pathological Complete Response (pCR), Disease-Free-Survival (DFS) and OS rates. A meta-analysis by Gao, et al., showed that CD4+ and CD8+ were positive predictors for long-term prognosis, while FOXP3 T cell infiltration was found to predict the worst prognosis [24].

Based on this evidence, immunotherapy—an approach that achieves its function by stimulating and regulating the host’s immune system and breaking tumour immunosuppression—has been studied in advanced TNBC. Notably, different possible biomarkers have been evaluated that can predict responses to immunotherapy: programmed cell death protein 1/programmed death-ligand 1 (PD-1/PD-L1) immune checkpoint, tumour mutational burden (TMB), defects in DNA mismatch repair proteins (dMMR) and a subsequent microsatellite instability-high (MSI-H). Table 1 summarises the most relevant results and the most important ongoing trials regarding immunotherapies.

#### 2.2.1. PD-1/PD-L1 Immune Checkpoint

Programmed death-ligand 1 (PD-L1) is expressed in approximately 20% of TNBC, and its interactions with PD-1, a transmembrane receptor protein on the surface of cells in the adaptive immune system, induce T cell inhibition and cancer immune system escape [25]. High expression of PD-L1 is significantly related to high TILs levels; on the other hand, the association between PD-L1 expression and prognosis is controversial. A meta-analysis by Lotfinejad, et al., showed no significant association between PD-L1 expression, tumour size and stage, and no correlation with OS and DFS in TNBC [26].

PD-L1 expression can be investigated using different scoring methods in immunohistochemistry: it can be assessed on immune cells (IC) only, on tumour cells (TC) only or on both IC and TC, as in the Combined Positive Score (CPS). In addition, each drug targeting PD-1/PD-L1 has a companion diagnostic test using a different method for scoring calculation, and there is no consensus on the cut-off value. Thus, in different clinical trials regarding immunotherapies, the PD-L1-positive population has been selected using different methods. In a post hoc exploratory sub-study of the IMpassion130 trial (first-line Atezolizumab plus Nab Paclitaxel for metastatic TNBC), Rugo, et al., investigated analytical concordance and clinical utility among different assays, to evaluate PD-L1 (the VENTANA SP142, VENTANA SP263 and Dako 22C3) with an unsatisfying overall percentage agreement. In fact, the analytical concordance for IC ≥ 1%, between SP142 and SP263 or 22C3 (CPS), was 69.2% and 68.7%, respectively [27]. Different assays identify populations that cannot be overlapped. In order to find the best selection of patients that may benefit from immunotherapy, there is an urgent need to harmonise PD-L1 evaluation, finding the best assay for the right drug [28].

Monoclonal antibodies against PD-1 and PD-L1 can lead to an immune-mediated response against the tumour; several clinical trials are evaluating the role of immunotherapy either in early-stage TNBC or in the metastatic setting.

In the IMPassion130 trial, patients were randomised to receive first-line nab-paclitaxel with the anti-PD-L1 atezolizumab or a placebo, until progression or unacceptable toxicity. Co-primary endpoints were PFS in the intention-to-treat (ITT) and in PD-L1-positive (defined as IC covering ≥ 1% of the tumour area as determined by VENTANA SP142 PD-L1 assay) populations, and OS tested hierarchically in the ITT population and, if significant, in the PD-L1 IC-positive population. Progression-free survival was longer in the atezolizumab–nab-paclitaxel group than in the placebo–nab-paclitaxel group in both the ITT population (median 7.2 months vs. 5.5 months, HR 0.8) and the PD-L1-positive subgroup (7.5 vs. 5.0 months, HR 0.62). However, the final OS analysis showed a statistically significant OS improvement in the PD-L1 positive population, with a median increase of 7.5 months (25.4 vs. 17.9 months, HR = 0.67), whereas OS in the ITT populations did not reach statistical significance [14].

On the other hand, IMPassion131, a clinical trial which evaluated atezolizumab-plus-paclitaxel as first-line treatment for metastatic TNBC, did not meet the primary endpoint of improvement in PFS in the PD-L1-positive population (IC ≥ 1% as determined by VENTANA SP142 PD-L1 assay). Similarly, there was no evidence of treatment benefit in PFS in the ITT population, and the addition of atezolizumab to paclitaxel did not improve OS (secondary endpoint) in either the PD-L1-positive or the ITT population [15].

Several hypotheses have been made as to why two similarly designed trials showed different results in terms of PFS and OS. The study populations were very similar according to disease setting, metastatic sites, PD-L1 expression and previous chemotherapy with taxanes and anthracyclines. Different clinical activity of paclitaxel and nab-paclitaxel has already been investigated in the GeparSepto-GBG69 trial, in which nab-paclitaxel increased the pathological complete response rate in neoadjuvant settings, compared to paclitaxel [29].

In addition to the potentially different efficacy of the chemotherapy backbone, the impact of steroids premedication associated with paclitaxel could be considered as a factor potentially reducing immunotherapy activity.

This hypothesis is undermined by observations in KEYNOTE-355, which described the benefit of immunotherapy both in paclitaxel and nab-paclitaxel subgroups—noting, however, that the trial was not designed to investigate differences in efficacy between different CT schedules [16,17].

Considering the heterogeneity of the immune microenvironment in TNBC, several aspects could confer distinct responses to immunotherapy, such as different TILs levels within the studied populations, tumour mutational burden and immune gene signatures [30].

Based on the progression-free survival results from the IMpassion130 trial for people with metastatic triple-negative breast cancer whose tumours expressed PD-L1 (≥1%), in 2019 the FDA granted accelerated approval for the combination of atezolizumab and nab-paclitaxel as frontline treatment of PD-L1+ mTNBC. Furthermore, the EMA approved atezolizumab in combination with nab-paclitaxel in the same setting. However, the FDA withdrew accelerated approval for atezolizumab, after negative OS results in the ITT population of IMPassion131. Thus, atezolizumab is no longer available for the treatment of PD-L1+ mTNBC in the United States, while it is still available in Europe.

Pembrolizumab (an anti-PD1 agent) activity in TNBC was assessed in phase II trial Keynote086, where it was administered as a monotherapy in previously treated (cohort A) and untreated (cohort B, PD-L1-positive defined as CPS ≥ 1 using the IHC 22C3 PDL-1 assay) mTNBC. In the untreated subgroup, median PFS was 2.1 months, median OS was 18 months and the ORR was 21.4%; no grade 4 AEs were described. In cohort A, 61.8% of enrolled patients were PD-L1 positive; the ORR was 5.3 in the total population and 5.7 in the PD-L1-positive subgroup. The disease control rate was 7.6% and 9.7%, respectively. Median PFS was 2.0 months, and the 6-month rate was 14.9%. Median OS was 9.0 months, and the 6-month rate was 69.1% [19].

KEYNOTE-355 showed that first-line pembrolizumab plus chemotherapy (nab-paclitaxel; paclitaxel; or gemcitabine/carboplatin) improved PFS and ORR, versus chemotherapy plus a placebo, in the patients with TNBC cancers and CPS ≥ 10, using the IHC 22C3 PD-L1 assay. In this subset of patients, median PFS was 9.7 months in the pembrolizumab-plus-chemotherapy arm, and 5.6 months in the placebo arm (HR 0.65); median overall survival was 23.0 months in the experimental arm vs. 16.1 months in the control arm (HR 0.73). No benefit was reported for the addition of pembrolizumab by applying a cut-off of CPS ≥ 1 or CPS > 0 [16,17].

Pembrolizumab did not significantly improve OS as monotherapy in previously treated mTNBC versus chemotherapy, in Keynote-119: median OS in patients with CPS > 10 was 12.7 in the pembrolizumab group versus 11.6 in the chemotherapy arm (HR 0.78). In patients with CPS of 1 or more, median OS was 10.7 in the experimental arm and 10.2 in the control arm. In the intent-to-treat population, median OS was 9.9 months for the pembrolizumab group and 10.8 for the chemotherapy group [18].

Pembrolizumab is now approved by the FDA and the EMA for the treatment of locally recurrent unresectable or metastatic TNBC in adults whose tumours express PD-L1 with a combined positive score (CPS) ≥ 10, and who have not received prior chemotherapy for metastatic disease.

Further investigations could be necessary, to evaluate combinations between two immunotherapy drugs (anti-PD-1/anti-CTLA4), considering the immunogenic substrate of triple-negative breast cancer, and the benefit in overall survival we observed with immunotherapy combinations in other diseases (melanoma, renal cell carcinoma, lung cancer). 

The NCT02536794 single-arm trial tested anti-PD-1 durvalumab in combination with anti-CTLA4 tremelimumab, in both TNBC and oestrogen receptor-positive BC. The ORR was 43% in TNBC, while median OS and median PFS were not reached in the same subgroup; the study was discontinued due to the ORRs not meeting the required criteria to move on to the next stage [20].

SYNERGY, an ongoing phase I/II trial, is testing the combination of anti-PD-L1 durvalumab, anti-CD73 oleclumab and chemotherapy (carboplatin plus paclitaxel), as first-line treatment in mTNBC (clinicaltrials.gov identifier NCT036 16886).

Concerning further immunotherapy combinations, leramilimab (LAG525), an anti-LAG3 antibody, when combined with spartalizumab, an anti-PD-1 antibody, led to durable RECIST responses in a phase I/II trial, including 2 of 5 TNBC patients [21].

#### 2.2.2. TMB and MSI-H/dMMR

Tumour mutational burden (TMB) refers to the number of somatic mutations per megabase of DNA measured using whole exome or gene panel sequencing. A high TMB, defined as ≥10 mut/Mb, has been associated with a high number of new antigens on the cell surface, and this highly immunogenic microenvironment is susceptible to immunotherapy in different types of solid tumours [31]. There is limited data regarding the role of TMB in MBC. A high TMB is present in about 5–11% of MBC, predominantly TNBC, and this finding is also associated with high TILs and BRCA1/2 germline mutations [31]. These features seem to predict good sensitivity to checkpoint inhibitors; however, no difference in OS has been observed in MBC patients with high TMB, treated or not with immunotherapy [32,33,34].

Defects in DNA mismatch repair proteins (dMMR) and a subsequent microsatellite instability-high (MSI-H), leads to dysfunction in DNA replication, promotion of mutations leading to oncogenesis, and stimulation of the tumour immune microenvironment. The frequency of MSI-H/dMMR in BC is estimated below 2%, and dMMR BC are frequently TNBC, high grade and associated with high TILs, but their use as a prognostic biomarker is unclear [35,36].

Nevertheless, based on the results of KEYNOTE-158, which demonstrated the significant clinical benefit of pembrolizumab among patients with previously treated advanced MSI-H/dMMR non-colorectal cancer (including MBC), in June 2020 the FDA approved pembrolizumab for any advanced solid tumours with MSI-high status, dMMR, or high TMB which have progressed following prior therapy or without alternative treatment options (tumour agnostic approval); meanwhile, the EMA has not yet approved that indication. Therefore pembrolizumab may represent a therapeutic option also for metastatic TNBC with these features (TMB high or MSI-H/dMMR) [22].

## 3. Targeting Intracellular Pathways

### 3.1. PolyADP Ribose Polymerases (PARP) in BRCA1/2 Mutated Patients and TNBC

PolyADP ribose polymerases are a large family of multifunctional enzymes with an important role in DNA repair mechanisms and genome integrity. PARP1 is the most important member of the family, and it is essential for single-strand breaks (SSB) repair through the base excision repair pathway [37]. Breast cancer cells affected by mutation in BRCA1/2 genes are deficient in the DNA for double-strand breaks (DSB) repair. Indeed, both BRCA1 and BRCA 2 are key components in homologous recombination repair (HRR), the major pathway to repairing the double-strand damage [38].

The PARP inhibitor (PARPi) prevents the repair of SSB, which seem to be converted to DSB. As cells with BRCA1 are deficient in HRR, the accumulation of DSB leads to cell apoptosis and death. Several trials have evaluated the use of PARP inhibitors, in different settings, in patients with BRCA1/2 mutations.

OlympiAD Trial is a phase III study which enrolled MBC patients with both triple-negative and HR+/HER2- tumours carrying a BRCA1/2 germline mutation, who had received a maximum of two lines of chemotherapy, to receive the PARPi olaparib or physician’s choice single-agent chemotherapy. In the primary analysis, PFS was longer in the olaparib arm (7 vs. 4.2 months, HR0.58), and the benefit was confirmed also in the TNBC subgroup. On the other hand, there was no statistically significant difference in OS between the two arms; the extended follow-up confirmed the previous results; however, patients who had not received any prior treatment for metastatic setting experienced an OS benefit of 7.9 months [39].

The EMBRACA trial was an open-label, randomised, phase III trial comparing the efficacy and safety of the PARPi talazoparib with a protocol-specified, single-agent therapy (capecitabine, eribulin, gemcitabine or vinorelbine) in germline BRCA mutated MBC patients. The median progression-free survival among patients in the talazoparib group was longer (8.6 months vs. 5.6 months, HR 0.54). The benefit in PFS was confirmed in all subgroups, even in patients with TNBC. Talazoparib showed a better response rate and longer durations of response; however, in the final OS analysis, talazoparib did not improve overall survival over chemotherapy [40].

Thus, in these studies, PARP inhibitors showed improvement in PFS and response rate, compared with standard chemotherapy, in the overall population and in the TNBC subgroup, but no benefit in OS. This could be explained by considering the impact of subsequent treatments and the crossover rate. Moreover, the OlympiAD trial was not empowered to detect OS differences between the two arms.

Based on the results of these trials, olaparib and talazoparib are now approved by the FDA and the EMA, for triple-negative metastatic breast cancer patients harbouring BRCA 1 or 2 mutations previously treated with chemotherapy before or after surgery.

The association between PARP inhibitors and immunotherapy, chemotherapy or targeted therapy is under evaluation as well.

The single-arm phase I/II trial, TOPACIO/Keynote-162, investigated the efficacy and safety of PARPi niraparib combined with pembrolizumab in patients with TNBC: it showed promising results, confirming the synergic activity of PARPi and immunotherapy previously evaluated in pre-clinical models [41]. Similar results were obtained in the MEDIOLA trial, a phase I/II basket study investigating niraparib plus durvalumab in patients with germline BRCA mutation [42].

Several other clinical trials, aimed at evaluating the safety and efficacy of PARP inhibitors in combination with immune checkpoint inhibitors, are ongoing. The DORA trial is a phase II multicentre study aiming to assess efficacy in terms of PFS of olaparib in combination with durvalumab in platinum-treated patients with TNBC (clinicaltrials.gov identifier NCT03167619). Medi4736 is a phase I/II study evaluating durvalumab in combination with olaparib and/or cediranib for advanced solid tumours, including ovarian cancer, TNBC, lung cancer, prostatic cancer and colorectal cancer (clinicaltrials.gov identifier NCT02484404). NCT02849496 is a phase II trial testing olaparib alone or in combination with atezolizumab in Her2-negative breast cancer (clinicaltrials.gov identifier NCT02849496). NCT04683679 is a phase II trial evaluating pembrolizumab and ablative radiotherapy with or without olaparib in TNBC; enrolled patients must have at least one tumour site for which palliative RT is considered appropriate, and they must not carry any BRCA mutation (clinicaltrials.gov identifier NCT04683679). Javelin BRCA/ATM is a phase II single-arm trial evaluating objective response with avelumab plus talazoparib, in patients with solid tumours harbouring BRCA or ATM mutations (clinicaltrials.gov Identifier NCT03565991). The NCT04690855 TARA trial is a phase II study investigating the combination of talazoparib, atezolizumab and radiotherapy in patients with TNBC, PD-L1 positive and BRCA negative (clinicaltrials.gov identifier NCT04690855). TALAVE is a phase I/II study investigating the safety and efficacy of induction with talazoparib followed by the combination of talazoparib and avelumab in advanced breast cancer, regardless of tumour biological features (clinicaltrials.gov identifier NCT03964532).

Brocade3 is a phase III trial that showed an improvement in median PFS in patients with BRCA 1/2 mutations treated with carboplatin plus paclitaxel and the PARPi Veliparib, compared with the same chemotherapy regimen and placebo (16.6 months vs. 14.1 months). Median PFS in the TNBC group was 16.6 months in the veliparib-plus-carboplatin-paclitaxel arm vs. 14.1 months in the placebo-plus-carboplatin-paclitaxel arm (HR 0.72). In the same subgroup, the trial also showed an OS improvement in the experimental arm (35 vs. 30, HR 0.69) [43].

SWOG S1416 is a phase II study that randomised patients with TNBC, divided into three subgroups (gBRCA+, BRCA-like and non-BRCA-like) to receive cisplatin ± PARPi veliparib; the primary endpoint was PFS in all subgroups of patients. In the BRCA+ subgroups, PFS was not statistically significant, but numerically better (HR 0.64, *p* 0.14). In the BRCA-like group, improved PFS was noticed in the cisplatin + velaparib arm (5.7 vs. 4.3, HR 0.58, *p* 0.023). Non-BRCA-like patients did not show any benefit from the combination [44].

Furthermore, recent studies have demonstrated that PARP inhibitors could play an important role as a maintenance treatment of advanced ovarian cancer, with improved PFS in patients with CR or PR, compared to platinum-based chemotherapy. Relying on this data, PARP inhibitors seem to deserve further investigation, in order to understand their role as a form of maintenance therapy in TNBC [45].

### 3.2. Androgen Receptor (AR)

The androgen receptor (AR) belongs to the steroid receptor family, and acts as a nuclear transcription factor. Upon ligand binding, the AR translocates from the cytoplasm to the nucleus, where it binds to specific sequences of DNA and promotes cell proliferation [46]. The androgen receptor pathway is emerging as a potential therapeutic target in breast cancer, and AR-targeted treatment for breast cancer is an area of active investigation. AR positivity, defined as AR ≥ 10%, accounts for approximately 30–35% of TNBC, and is associated with the LAR subtype, low tumour grade, lower risk of nodal involvement, and older age at diagnosis [47]. It also has a prognostic value because patients with AR-positive tumours have significantly longer metastatic intervals and overall survival [48,49]. 

Several trials have explored the role of different agents able to interfere with the AR-pathway in metastatic TNBC, notably: abiraterone acetate, an oral agent that inhibits androgen biosynthesis by inhibiting 17α-hydroxylase/C17,20-lyase (CYP17); bicalutamide, a non-steroidal anti-androgen; and enzalutamide, a potent oral androgen receptor inhibitor.

The UCBG 12-1 study—a phase II trial of abiraterone acetate plus prednisone in patients with pre-treated triple-negative androgen receptor-positive locally advanced or MBC—showed a 6-month clinical benefit rate (CBR) of 20.0% and an ORR of 6.7%, with a median PFS of 2.8 months [50].

In a phase II trial exploring bicalutamide (150 mg daily) in 26 AR-positive, oestrogen receptor-negative MBC and progesterone receptor-negative MBC, although there were no complete or partial responses, two patients had stable disease for up to 6 months, and an additional five patients had stable disease for more than 6 months, with a CBR of 19% [51].

Similar results were observed in a phase II trial of enzalutamide in patients with AR-positive TNBC, with a CBR of 28% [52].

These data suggest a role for anti-androgen agents in the treatment of locally advanced or MTNBC expressing the AR. Additionally, there are several ongoing clinical trials in the metastatic setting, to evaluate the use of the AR blockade in combination with various targeted therapies, including PI3K inhibitors, based on the cross-talk between the AR pathway and several other key signalling pathways, including the PI3K/Akt/mTOR and MAPK pathways [46].

Results from a phase Ib/II study (TBCRC032) investigating the safety and efficacy of enzalutamide alone, or in combination with the PI3K inhibitor taselisib in patients with metastatic AR+ (≥10%) breast cancer, showed that the combination of enzalutamide and taselisib increased CBR in TNBC patients with AR+ tumours [53].

Further studies are therefore needed, to explore new combinations in AR+ TNBC, like the ongoing NCT03090165 phase I/II trial evaluating the safety and efficacy of ribociclib (a CDK4/6 inhibitor) in combination with bicalutamide in advanced AR+ TNBC (ClinicalTrials.gov Identifier: NCT03090165).

### 3.3. PI3K/AKT/mTOR

The phosphoinositide 3 kinase (PI3K)/Akt/mammalian target of rapamycin (mTOR) pathway is a complicated signalling pathway involved in tumour growth and proliferation. The PIK3CA gene encodes a subunit of class IA PI3K that is essential for phosphatidylinositol 4,5 bisphosphate (PIP2) phosphorylation and conversion to phosphatidylinositol 3,4,4-triphosphate (PIP3). PIP3 acts as a second messenger activating AKT, a serine/threonine kinase whose activation triggers signalling through a multitude of AKT downstream targets that control cell survival, growth, proliferation and apoptosis. mTOR is a serine/threonine protein kinase, which is found downstream of PI3K and Akt [54,55]. Conversely, phosphatase and tensin homolog deleted on chromosome ten (PTEN) is a tumour suppressor that reduces activation of AKT by hydrolysing PIP3 to PIP2. The PI3K/AKT/mTOR signalling pathway is often activated in triple-negative breast cancer through activating mutation in PIK3CA and AKT1 or inactivating alterations in PTEN. All these alterations occur in about 25% of TNBC [56].

A study by Hu, Zhu, Zhong, et al., showed that E545K and H1047R PIK3CA mutations confer a more aggressive phenotype in TNBC, and resistance to adjuvant chemotherapy, and that loss of PTEN confers resistance to PD-L1 blockade [57].

Several trials have evaluated drugs targeting the PI3K/AKT/mTOR pathway in TNBC.

The PAKT trial is a phase II trial that randomises patients with untreated metastatic TNBC, to receive paclitaxel with the AKT inhibitor capivasertib or a placebo. The PFS median (the primary endpoint) was 5.9 months with capivasertib, and 4.2 with the placebo; in patients with PIK3CA/AKT/PTEN, the altered tumours PFS median was 9.3 months with capivasertib vs. 3.7 months with the placebo (HR 0.30). Thus, the addition of the AKT inhibitor capivasertib resulted in significantly longer PFS, and the benefit was more pronounced in patients with PIK3CA/AKT1/PTEN-altered tumours. The final results showed a numerical improvement in OS in the capivasertib arm (19.1 vs. 13.5 months; HR 0.70) in the intent-to-treat population, but no statistically significant difference; similarly, no difference was shown in terms of clinical benefit between patients with or without PI3K/AKT/mTOR mutations, as median OS was numerically better but not statistically significant in the capivasertib arm in comparison to the control arm in both subgroups [58,59].

The LOTUS trial is a double-blind placebo-controlled randomised phase II trial evaluating the oral AKT inhibitor, ipatasertib. Patients were randomised to receive paclitaxel plus ipatasertib or a placebo. The co-primary endpoints were progression-free survival in the intention-to-treat population and progression-free survival in the PTEN-low (by immunohistochemistry) population. In the final analysis, an improvement in PFS was observed both in the ITT population and in the PTEN-low subgroup (6.2 vs. 4.9, HR 0.60 in the ITT population; 6.2 vs. 3.7 months, HR 0.59 in the PTEN-low subgroup). Further analyses in the subgroup of 42 patients with PI3K/AKT/mTOR mutations showed median PFS of 9.0 months in the experimental arm versus 4.9 months in the placebo arm. Median OS was numerically longer in ipatasertib + paclitaxel (25.8 months with 16.9 months; HR 0.80) in all biomarkers-defined subgroups, with no enhanced efficacy in patients carrying PI3K/AKT/mTOR mutations [60,61].

Unfortunately, in contrast to the results of the LOTUS trial, the phase III ipatunity130 trial did not show PFS improvement with the addition of the AKT-inhibitor ipatasertib to paclitaxel in TNBC with PIK3CA/AKT1/PTEN mutations [62].

Clinical trials are ongoing to further evaluate these drugs, notably their combination with other chemotherapies or immunotherapy.

NCT04464174 PathFinder is a phase II study investigating the safety (primary endpoint) and PFS, ORR and OS (secondary endpoints) of ipatasertib in combinations with carboplatin, eribulin or capecitabine, in patients with unresectable or metastatic TNBC (ClinicalTrials.gov Identifier: NCT04464174). NCT03742102 Begonia is a phase Ib/II stage, open-label, multicentre study to determine the efficacy and safety of durvalumab in combination with novel oncology therapies with or without paclitaxel and durvalumab + paclitaxel as a first-line treatment in patients with TNBC; Experimental arm 2 is investigating capivasertib in association with paclitaxel and durvalumab (Clinicaltrials.gov Identifier: NCT 03742102). CAPitello290 is a phase III ongoing trial evaluating the efficacy and safety of capivasertib in combination with paclitaxel in first-line treatment of patients with metastatic TNBC in an unselected population (ClinicalTrials.gov Identifier: NCT03997123). NCT04177108—IpaTunity170 is a phase III, double-blind, placebo-controlled trial investigating ipatasertib in combination with atezolizumab and paclitaxel (ClinicalTrials.gov Identifier: NCT04177108).

Several trials are also evaluating the potential role of PI3Ks inhibitors in TNBC.

A study by Zhi, Zhu, Getzenberg, et al., demonstrated that upregulation of the PIK3/AKT/mTor pathway seems to be associated with resistance to multiple chemotherapy drugs, especially to microtubules targeting drugs in prostatic cancer [63]. Relying on these data, the PI3K-inhibitor alpelisib has been tested in association with paclitaxel in a phase Ib trial including different solid tumours. However, the study has been interrupted for high-grade toxicities in terms of hyperglycemia and diarrhoea [64].

A phase Ib/II trial enrolled a small number of patients with HER2 negative breast cancer to investigate the safety and efficacy of nab paclitaxel + alpelisib; the results showed promising efficacy, especially in PI3K mutated tumours. ER+ and TNBC had similar benefits in efficacy, with an objective response rate of 60% and 58%, respectively [65].

EPIK-B3 (NCT04251533) is a phase III randomised, double-blind, placebo-controlled ongoing trial evaluating the efficacy and safety of alpelisib in combination with nab-paclitaxel in patients carrying PIK3CA mutations or PTEN-Loss (ClinicalTrials.gov Identifier: NCT04251533).

As PIK3CA mutations are found to be more frequent in TNBC-expressing androgen receptors (40% of AR+ tumours vs. 4% in AR- TNBC), different trials have also evaluated the safety and efficacy of enzalutamide in association with the PI3K inhibitors alpelisib or taselisib [66].

TBCRB IB/II is a multicentre study which demonstrated that enzalutamide has a good safety profile in association with taselisib, and that it seems to increase clinical benefit when compared to enzalutamide alone in TNBC: the clinical benefit rate was 35.7% for patients receiving the combinations. There was no significant difference in clinical benefit in the PI3K/AKT/mTOR mutated subgroup (42.9% vs. 28.6 %, *p* = 1); patients with the LAR subtype trended to have a higher benefit (75% vs. 12.5%, *p* = 0.06) [53].

NCT03207529 is an ongoing phase I trial investigating side effects and best dosage of alpelisib plus enzalutamide in patients with PTEN-positive MBC-expressing androgen receptors, including TNBC (ClinicalTrials.gov Identifier: NCT03207529).

In vitro and in vivo studies have shown that TNBC-expressing luminal androgen receptors were sensitive to CDK4/6 inhibition; in particular, sensibility to cyclin-dependent kinase inhibitors seemed to be related to CDK2 expression [67]. Relying on these data, a phase IB trial (PIPA) evaluated the outcomes of treatment with taselisib + palbociclib in various solid tumours, including a cohort of TNBC. Among patients with BC not expressing hormone receptors (8 HER-2 negative and 3 HER-2 positive), CBR was 27%, and median PFS was 4.3 months, but the ORR was 0% (0/11) [68].

Finally, PI3K inhibitors have also been involved in clinical trials evaluating the responses to combinations with immunotherapy and chemotherapy, such as MARIO3, a phase II clinical trial evaluating the addition of the PI3K inhibitor eganelisib to nab-paclitaxel and atezolizumab. Recent results, presented at the 2022 San Antonio breast cancer symposium, showed promising anti-tumour activity (ORR 55.3% and DCR 84.2%) with a good safety profile for this novel triplet regimen. The responses were seen to be irrespective of PD-L1 status: the ORR was 66.7% in PD-L1 positive and 47.8% in PD-L1 negative patients. The disease control rate was 91.7% in PD-L1 positive and 78.3% in PD-L1 negative patients [69].

ARC-2 (NCT03719326) is a Phase I/Ib, open-label, dose-escalation and dose-expansion study, evaluating the safety, tolerability, and clinical activity of adenosine receptor inhibitor etrumadenant + pegylated liposomal doxorubicin (PLD) ± eganelisib in patients with locally advanced or metastatic TNBC or ovarian cancer. The triplet was well tolerated without evidence of addictive toxicity; both doublet and triplet combination regimens were associated with clinical benefit. Of 18 evaluable patients treated with the doublet regimen, 2 patients (1 TNBC) achieved a partial response, and 9 had stable disease. Of 12 evaluable patients treated with the triplet regimen, 1 patient achieved a complete response (ovarian), 4 had a partial response (2 ovarian, 2 TNBC) and 4 had stable disease as a best response [70].

Overall, the outcomes of these studies are improving the knowledge of the impact of PI3K pathway inhibition on metastatic TNBC treatment and, in future, they will allow us to improve the treatment options (considering both targeted therapy alone or in combination with chemotherapy/immunotherapy).

### 3.4. NOTCH

The Notch receptor is a single-pass transmembrane protein. The NOTCH signalling pathway activates many genes associated with cell differentiation, proliferation and cell death. There are 26 NOTCH gain-of-function mutations present in approximately 10% of TNBC, and they represent a poor prognostic factor associated with decreased survival [71,72]. NOTCH signalling is activated via cleavage of the transmembrane protein to release the NOTCH intracellular domain, which transports products to the nuclease, acting as a transcription factor at this level.

NOTCH inhibitors have been developed to target this pathway. In a phase Ib study evaluating PF-03084014, a selective gamma-secretase inhibitor, in combination with docetaxel in patients with MTNBC, 16% of patients had a partial response in first-line treatment of advanced disease, and 36% of patients achieved stable disease as best response; the 6 months-PFS rate was 17.1% [73].

Further studies investigating the use of NOTCH inhibitors in mTNBC are therefore under development, such as the TENACITY trial evaluating pan-NOTCH gamma-secretase inhibitor AL101 monotherapy (ClinicalTrials.gov Identifier: NCT04461600).

### 3.5. NTRK

Chromosomal translocations involving the neurotrophin receptor tyrosine kinase genes NTRK1, NTRK2 or NTRK3 lead to TRK gene fusions, resulting in the overexpression of the proteins, their constitutive activation and subsequent tumour growth. These events occur in approximately 1% of all solid tumours and <1% of BC [74]. 

A phase I–II trial evaluated the efficacy of larotrectinib, a tropomyosin receptor kinase inhibitor, in TRK fusion-positive patients affected by different solid tumours (only one patient had breast cancer), showing a 1-year ORR of 71%, durable anti-tumour activity and a manageable safety profile [75].

More recently, an integrated interim analysis of three phase I–II clinical trials (ALKA-372–001, STARTRK-1 and STARTRK-2), evaluating the anti-tumour activity and safety of entrectinib (another tropomyosin receptor kinase inhibitor) for patients with TRK-fusion-positive solid tumours, showed an ORR of 57% (7% complete response and 50% partial response) with a median DOR of 10 months. In this study, six patients (11%) had breast cancer [76].

Based on the results of these trials, larotrectinib and entrectinib received FDA and EMA approval for patients with TRK gene fusions, representing a highly effective treatment for these patients, irrespective of the tumour subtype. Thus, although TRK gene fusions are extremely rare in breast cancer, these drugs represent a possible therapeutic option for the subgroup characterised by this molecular alteration. 

Long-term results from ongoing phase II basket studies, like the STARTRK-2 trial (evaluating entrectinib for the treatment of patients with solid tumours harbouring NTRK 1/2/3, ROS1 or ALK Gene Rearrangements) and the NAVIGATE trial (investigating larotrectinib in adults and children with NTRK-fusion-positive solid tumours) will provide further data about the role of these NTRK inhibitors.

Table 2 summarises the most relevant results and the most important ongoing trials regarding therapies targeting intracellular pathways.

### 3.6. Glutaminase Inhibitors

The amino acid glutamine represents a source of energy used by cancer cells to support their rapid growth and proliferation. The intracellular processing of glutamine begins with its conversion to glutamate by the mitochondrial enzyme glutaminase, which can be considered as a therapeutic target for small molecules inhibitors ([77]). CB-839, an oral highly selective inhibitor of glutaminase, has been tested in association with paclitaxel in a phase I trial enrolling metastatic TNBC patients, including those previously treated with taxanes. At CB-839 dosage of 600 mg BID, the ORR was 22%, and the DCR was 47%. Patients with African ancestry showed the highest ORR (36%) and DCR (55%) ([78]). CX-839-007 is a phase II study further testing the activity of CB-839 + paclitaxel, both in untreated (1L) and treated, with two or more prior lines of systemic therapies (3L), TNBC patients. The primary endpoint was the ORR, and it was 41% in 1LmTNBC, and 12% in 3LmTNBC; in the two subgroups, the DCR was 86% and 36% ([79]). Considering the high glutamine utilization observed in TNBC, especially in patients with African ancestry, further investigations and trials are necessary, to better define the role of glutaminase inhibitors in TNBC.

**Table 2 cancers-14-03729-t002:** Intracellular pathways: summary of clinical trials.

Target	Relevance in TNBC	Drugs	Clinical Trials (Phase)	Outcomes	Indication Approved/Not Approved
PolyADP	10–20%	olaparib	OlympiAD (phase III) (olaparib vs. chemotherapy) [39]	PFS: 7 vs. 4.2 months	Approved by FDA and EMA
			DORA trial (phase II) NCT03167619 (olaparib + durvalumab vs. olaparib alone)	Efficacy assessed by PFS	Ongoing
			Medi4736 NCT02484404 (durvalumab + olaparib and/or cediranib)	ORR in ovarian cancer subgroup, recommended second dose	Ongoing
			NCT02849496 (phase II) (olaparib + atezolizumab vs. olaparib alone)	PFS	Ongoing
			NCT04683679 (RT + pembrolizumab + olaparib vs. RT+ pembrolizumab alone) (phase II)	ORR	Ongoing
		talazoparib	EMBRACA (phase III) (talazoparib vs. chemotherapy) [40]	PFS: 8.6 vs. 5.6 months	Approved by FDA and EMA
			Javelin BRCA/ATM (phase II) (avelumab + talazoparib, single arm)	ORR	Ongoing
			NCT04690855 TARA (phase II) (talazoparib + atezolizumab + RT single arm)	ORR	Ongoing
			NCT03964532 TALAVE (phase I/II) (talazoparib + avelumab, single arm)	AEs	Ongoing
		niraparib	TOPACIO-Keynote 162 (phase I/II) (niraparib + pembrolizumab) [41]	ORR: 21% in full population	Not approved
			MEDIOLA (phase I/II) (niraparib + durvalumab) [42]	12 weeks DCR: 80%, safety and tolerability: 11% grade 3 or more AEs	Not approved
		veliparib	BROCADE 3 (phase III) (veliparib + carboplatin+ paclitaxel vs. placebo+ carboplatin + paclitaxel) [43]	PFS: 16.6 vs. 14.1 months OS: 35 vs. 30 months	Not approved
			SWOG S1416 (phase II) (veliparib + cisplatin vs placebo + cisplatin) [44]	PFS BRCA+: not statistically significant; BRCA-like: 5.7 vs. 4.3 months; BRCA-: no benefit	Not approved
AR	30–35%	abiraterone	UCBG 12-1 (phase II) (abiraterone acetate + prednisone in single arm) [50]	6 months CBR: 20% ORR: 6.7% PFS: 2.8 months	Not approved
		bicalutamide	TBCRC 011 (phase II) (bicalutamide in single arm) [51]	6 months CBR: 19%	Not approved
		enzalutamide	NCT01889238 (phase II) (enzalutamide in single arm) [52]	16 weeks CBR: 28%	Not approved
			TBCRC032 (phase IB/II) (enzalutamide + taselisib vs. enzalutamide) [53]	16 weeks CBR: 35.7% vs. 0%, 75% in TNBC LAR subtype with the combination	Not approved
			NCT03090165 (phase I/II) (bicalutamide + ribociclib in single arm)	16 weeks CBR	Ongoing
PI3K/AKT/mTOR	25%	capivasertib	PAKT (phase II) (paclitaxel + capivasertib vs. placebo) [58,59]	PFS: 5.9 vs. 4.2 months in the ITT populations; 9.3 vs. 3.7 months in PIK3CA/AKT/PTEN mutated population	Not approved
			CAPitello290 (phase III) (paclitaxel + capivasertib vs. paclitaxel + placebo)	OS	Ongoing
			NCT03742102 Begonia (phase Ib/II) (durvalumab, paclitaxel, capivasertib)	Safety	Ongoing
		ipatasertib	LOTUS (phase II) (paclitaxel + ipatasertib vs. paclitaxel + placebo) [60,61]	PFS: 6.2 vs. 4.9 months, PFS in PTEN-low: 6.2 vs. 3.7 months	Not approved
			Ipatunity130 (phase II) (paclitaxel + ipatasertib vs. pacltaxel + placebo) [62]	PFS: 9.3 months in both arms	Not approved
			NCT04464174 PathFinder (phase IIa) (ipatasertib + eribulin vs. ipatasertib + capecitabine vs. ipatasertib + carboplatin + gemcitabine)	safety and tolerability	Ongoing
			NCT04177108 IpaTunity170 (phase III) (paclitaxel ± atezolizumab ± ipatasertib)	PFS, OS	Ongoing
		alpelisib	NCT02051751 (phase I) (paclitaxel + alpelisib, dose finding study) [64]	Dose limiting toxicity, dose expansion: 41.7 % of populations experience dose limiting toxicities; dose expansion not initiated	Not approved
			NCT02379247 (phase I/II) (nab-paclitaxel + alpelisib) [65]	Recommended phase II dose of alpelisib, ORR of subject treated with phase II dose of alpelisib: 60% in ER+ population, 58% in TNBC population	Not approved
			EPIK-B3 NCT04251533 (phase III) (alpelisib + nab-paclitaxel)	PFS, ORR	Ongoing
		eganelisib	NCT03207529 (phase I) (alpelisib + enzalutamide, single arm)	MTD	Ongoing
			MARIO3 (phase II) NCT03961698 (eganelisib + nab-paclitaxel + atezolizumab) [69]	Complete response rate. Other anti-tumour activity data: ORR: 55.3%, DCR: 84.2% in ITT population, ORR: 66.7% in PD-L1-positive, 47.8% in PD-L1-negative patients. DCR: 91.7% in PD-L1-positive, 78.3% in PD-L1-negative patients	Not approved
		taselisib	ARC-2 (NCT03719326) (phase I/IB) (etrumadenant + pegylated liposomal doxorubicin (PLD) ± eganelisib) [70]	Safety and tolerability	Ongoing
			TBCRB (phase IB/II) (enzalutamide + taselisib vs. enzalutamide alone) [53]	16 weeks CBR: 35.7% vs. 0% in patients receiving the combination; 42.9% vs. 28.6% in PIK3CA/AKT/mTOR mutated population; 75% vs. 12.5% in LAR subtype	Not approved
			PIPA trial (phase I) (taselisib + palbociclib) [68]	Recommended dose for phase II, safety and toxicity. Other findings in ER negative population CBR: 27%, median PFS: 4.3 months	Not approved (trial status unknown)
NOTCH	10%	PF-03084014	NCT01876251 (phase IB) (PF-03084014 + docetaxel single arm) [73]	6 months PFS: 17.1% ORR: 16%	Not approved
		AL101	TENACITY NCT04461600 (phase II) (AL101 in single arm)	ORR	Ongoing
NTRK	<15	larotrectinib	NAVIGATE (phase I–II) (larotrectinib in single arm) [75]	1 year ORR 71%	Approved by FDA and EMA
		entrectinib	ALKA-372–001, STARTRK-1, and STARTRK-2 (phase I–II) (entrectinib in single arm) [76]	ORR: 57%, DoR: 10 months	Approved by FDA and EMA
Glutaminase		CB-839	Abstract PD3–13 (phase I) (CB-839 + paclitaxel) [78]	ORR 22% DCR 47%	Not approved
			NCT03057600 (phase II) (CB-839 + paclitaxel) [79]	ORR 41% (1L); 12% (3L) DCR 86% (1L); 36% (3L)	Not approved

## 4. Targeting Cancer by Exploiting a Cellular Target to Convey Chemotherapy to the Tumour: Antibody–Drug Conjugates (ADCs)

ADCs consist of a monoclonal antibody conjugated to a potent cytotoxin. The monoclonal antibody is directed against an antigen on the surface of the cancer cell. Several promising antigens have been identified in TNBC: low human epidermal growth factor receptor 2 (HER2) expression, trophoblast cell-surface antigen (Trop-2), the glycoprotein nonmetastatic b (GPNMB), LIV-1 and HER3 [38,39,40].

Table 3 summarises the most relevant results, and the most important ongoing trials regarding ADCs.

### 4.1. HER2-Low

HER2-positive BC is characterised by the overexpression and/or amplification of human epidermal growth factor receptor 2 (HER2), and accounts for approximately 15% of all BCs [88]. There is a group of BCs named HER2-low, defined by low HER2 protein expression but undetectable gene amplification (immunohistochemistry 1+ or immunohistochemistry 2+ with negative in situ hybridization), that represents approximately 45–55% of all BCs, and that has shown sensitivity to many novel anti-HER2 targeted agents [89]. Patients with HER2-low BC represent a heterogeneous group, consisting mainly of hormone-receptor-positive tumours (65–83%), but also of a group of HR-negative tumours. These subpopulations have different molecular profiles: the HER2-low HR+ are enriched with luminal subtypes, while the HR- are characterised by the predominance of the basal-like subtype, underlining prognostic differences within the group [90].

Trastuzumab deruxtecan (T-DXd, formerly DS-8201a) is a human epidermal growth factor receptor 2 (HER2)-targeted antibody–drug conjugate, with a topoisomerase I inhibitor payload. Pre-clinical studies have demonstrated that the released payload of T-DXd, unlike T-DM1, is cell-membrane-permeable, and that T-DXd induces a bystander cytotoxic effect on cells in close proximity to targeted HER2-expressing tumour cells, regardless of their HER2 status [91].

In a phase Ib study, evaluating T-DXd in patients with MBC and low HER2 expression, the investigator-reported confirmed ORR was 44.4%, with a median duration of response of 10.4 months, while median PFS and OS were 11.1 and 29.4 months, respectively, with a generally manageable safety profile [80].

These results led to the DESTINY-Breast04 phase III, randomised, multicentre trial, comparing the efficacy and safety of T-DXd 5.4 mg/kg versus physician’s choice chemotherapy (capecitabine, eribulin, gemcitabine, paclitaxel or nab-paclitaxel) in patients with HER2-low, unresectable and/or MBC. In the results, T-Dxd was statistically superior to physician’s choice therapy (TPC), both in terms of PFS (9.9 vs. 5.1 months, HR0.50, *p* < 0.0001) and OS (23.4 vs. 16.8 months, HR = 0.64, *p* = 0.0010). In an exploratory analysis conducted in HR- patients, median PFS was 8.5 and 2.9 months for TDx and TPC (HR = 0.46, 95%CI, 0.24–0.89), respectively, whereas median OS was 18.2 and 8.3 months, favouring TDx (HR = 0.48, 95%CI, 0.24–0.95) [81]. This novel ADC is also under investigation for mTNBC in a phase IB/II study, in combination with the immune checkpoint inhibitor durvalumab (BEGONIA trial, ClinicalTrials.gov Identifier: NCT03742102).

### 4.2. TROP-2

Trop-2 is a transmembrane glycoprotein involved in cell migration, cell proliferation and metastasisation. In a study that evaluated Trop-2 expression among different BC histological subtypes, its over-expression was detected in 62% of BC samples, reaching 78% in the TNBC subgroup [92].

Sacituzumab govitecan (IMMU-132) is an antibody targeting Trop-2, linked to the topoisomerase-I inhibitor SN-38, the active metabolite of irinotecan that induces DNA damage [82]. Based on the results of phase I/II clinical trial IMMU-132–01—which showed the efficacy of sacituzumab govitecan, with a 33.3% response rate and median duration of response of 7.7 months in heavily pre-treated MTNBC patients—in April 2020 the FDA granted accelerated approval to sacituzumab govitecan for patients with MTNBC who had received at least two prior therapies for metastatic disease.

The randomised phase III ASCENT trial evaluated sacituzumab govitecan versus physician’s choice chemotherapy (vinorelbine, capecitabine, eribulin or gemcitabine) in pre-treated (at least two previous chemotherapies) MTNBC. In this trial, sacituzumab govitecan prolonged median PFS (primary endpoint) and OS compared with physician’s choice chemotherapy (5.6 vs. 1.7 months and 12.1 vs. 6.7 months, respectively) [83].

Based on these results, the EMA has authorised the use of sacituzumab govitecan for MTNBC patients who have received two or more prior systemic therapies, including at least one for advanced disease. Sacituzumab govitecan was also included in the last updated ESMO guidelines, as the preferred treatment option for MTNBC after taxanes [93].

A pre-specified, exploratory biomarker analysis from the ASCENT trial, evaluating the association between tumour Trop-2 expression and germline BRCA1/2 mutation status with clinical outcomes, suggested that sacituzumab govitecan benefits patients with previously treated MTNBC expressing high/medium Trop-2 compared with standard-of-care single-agent chemotherapy, and regardless of germline BRCA1/2 mutation status [94].

Further studies are needed to fully elucidate the outcomes among these patient populations and those with low Trop-2 expression. Several trials evaluating sacituzumab govitecan in combination with other agents for the treatment of MTNBC are currently ongoing (SEASTAR trial, ClinicalTrials.gov Identifier: NCT03992131; MORPHEUS-TNBC trial, ClinicalTrials.gov Identifier: NCT03424005; ClinicalTrials.gov Identifier: NCT04468061).

Another anti-TROP2 ADC, datopotamab deruxtecan (Dato-DXd), is under investigation for advanced BC.

In a phase I trial investigating Dato-DXd in mTNBC pre-treated patients, 34% of them experienced a complete or partial response with a manageable toxicity profile [84].

Based on these results, further trials are ongoing: TROPION-Breast02 is a phase III study of Dato-DXd versus investigator’s choice chemotherapy in patients with metastatic TNBC who are not candidates for PD-1/PD-L1 inhibitor therapy (ClinicalTrials.gov Identifier: NCT05374512); the BEGONIA trial is a phase IB/II trial evaluating different combination treatments, among which is the association of durvalumab and Dato-DXd, for mTNBC (ClinicalTrials.gov Identifier: NCT03742102).

### 4.3. GPNMB

Glycoprotein-NMB (GPNMB) is a transmembrane protein involved in cell migration, invasion, angiogenesis, or epithelial–mesenchymal transition; it is highly over-expressed in TNBC (approximately 40%), representing a biomarker of poor prognosis [95]. Glembatumumab vedotin is an ADC that binds to GPNMB to deliver the potent microtubule inhibitor monomethyl auristatin E (MMAE).

The EMERGE trial, a phase II study investigating the activity of glembatumumab vedotin compared with investigator’s choice chemotherapy in heavily pre-treated MBC, showed an ORR of 18% vs. 0% in the subgroup of patients with MTNBC, increasing to 40% vs. 0% in GPNMB-over-expressing MTNBC [85]. 

However, another phase II trial (METRIC), that compared glembatumumab vedotin to capecitabine in pre-selected GPNMB-over-expressing MTNBC patients, failed to demonstrate improved PFS (primary endpoint), ORR or OS, leading to discontinuation of the development of this ADC [86].

### 4.4. LIV-1

LIV-1 is a zinc transporter protein downstream target of STAT3, implicated in cell adhesion and epithelial–mesenchymal transition, and expressed in 65% of MTNBC samples [96]. Ladiratuzumab vedotin (SGN–LIV1A) consists of a humanised antibody conjugated through a proteolytically cleavable linker to monomethyl auristatin E, a potent microtubule-disrupting agent [96].

Interim results from a phase I study evaluating the safety and activity of ladiratuzumab vedotin in LIV-1-positive pre-treated MBC showed, among the triple-negative patients, an ORR of 32% (PR rate of 21%), a DCR of 64% and a clinical benefit rate of 36%, demonstrating encouraging anti-tumour activity in heavily pre-treated TNMBC [97].

Based on these promising results, SGN-LIV1A, in combination with pembrolizumab, is currently under evaluation in a phase IB-II trial (ClinicalTrials.gov Identifier: NCT03310957).

### 4.5. HER3

HER3, a member of the HER family, is over-expressed in different solid tumours, and plays an important role in cell proliferation and metastatic spread, being associated with worse survival [98]. Its over-expression in breast cancer varies between 17.5% and 43%, based on the antibody used for its evaluation by immunohistochemistry (IHC) [99].

Patritumab deruxtecan is an ADC comprising a recombinant fully human anti-HER3 monoclonal antibody linked to a linker containing a potent topoisomerase I inhibitor. 

Results from a phase I–II study (NCT02980341) investigating patritumab deruxtecan in pre-treated patients with HER3-expressing metastatic breast cancer, showed encouraging efficacy and adequate safety profile. In the TNBC subpopulation, the ORR and the DCR were 22.6% and 56.6%, respectively, whereas the median DOR was 5.9 months [87].

Based on these results, patritumab deruxtecan is currently under evaluation in a phase II trial (ClinicalTrials.gov Identifier: NCT04699630).

## 5. Targeting Protein Degradation Space (PROTAC): A New Protein Degradation Technology

Proteolysis-targeting chimeric (PROTAC) technology is a recently developed endogenous protein degradation tool that can ubiquitinate the target proteins through the ubiquitin-proteasome system, to achieve an effect on tumour proliferation [100].

Preliminary results from a phase I–II trial (NCT04072952) evaluating ARV 471, an oral PROTAC that degrades ER in ER-positive breast cancer cells, alone or in combination with palbociclib, in 21 patients affected by metastatic ER+/HER2- breast cancer, showed a 6-months CBR of 42% with a relatively favourable safety profile (no treatment-related grade 3 of 4 adverse events were reported) [100].

Further studies will investigate these novel molecules in TNBC: a novel PROTAC C8, obtained by conjugating PARP1/2 inhibitor olaparib to the protein degrader KB02, can induce potent and specific degradation of PARP2 by recruiting DCAF16 E3 ligase for treatment of wild-type TNBC, representing a promising lead compound for the treatment of BRCA-wild-type TNBC [101].

## 6. Targeting Both Cancer Cells and Tumour Environment by Low Repeated Drug Doses: Metronomic Chemotherapy

Metronomic chemotherapy (mCHT) refers to the repeated administration of low doses of a chemotherapy agent to maintain prolonged and active plasma concentrations, and to provide a favourable toxicity profile [102]. It does not only have a direct anti-tumour effect, but could explicate its primary action on the tumour microenvironment, by inhibiting angiogenesis and promoting immune response. Although this approach has historically been reserved for mildly aggressive tumours, this multiple mechanism of action makes metronomic chemotherapy an interesting approach to use also against TNBC.

In a multicentre phase II trial (Victor-2) evaluating the activity and safety profile of the oral metronomic combination of vinorelbine (VNR) and capecitabine (CAPE) in 80 advanced HER2-negative breast cancer patients, the subgroup analysis of the TNBC population (about 35%) showed a CBR of 35.7%, with a median duration of CB of 11.3 months; median time to objective response and median PFS were 2.1 and 4.7 months, respectively, with similar clinical activity when the combination regimen was used in both first-line and second-line settings; furthermore, it was well-tolerated without significant severe adverse events [103].

Montagna, et al., in another phase II trial, tested the combination of a triple-drug oral metronomic chemotherapy consisting of vinorelbine, cyclophosphamide and capecitabine (VEX regimen) in 25 previously untreated TNBC patients: the ORR was 27% and the CBR was 50%; median time to progression (TTP) and median time to death were 6.4 and 18.4 months, respectively. The VEX regimen was relatively well-tolerated, with only 9% of grade 3 hand–foot syndrome (HFS) [104].

The Victor-6 study is a multicentre retrospective cohort study which collected data from 584 MBC patients who received mCHT. A recent analysis of the metastatic TNBC subgroup (16% of the population) showed an ORR and a DCR of 17.5% and 64.9%, respectively. The best ORRs and DCRs were observed in first-line settings (20.9% and 76.7%), whereas tumour response decreased proportionally in later lines. Median PFS and OS were 6.01 months and 12.1 months, respectively, and they were longer for capecitabine-based regimens than for CTX-based and vinorelbine-based ones. Moreover, longer PFS was observed when mCHT was used in the first-line setting than in the second and subsequent lines. Median OS was 18.2 months for capecitabine-based regimens and 11.8 months for vinorelbine- and CTX-based ones, while similar results were observed for OS according to the line of treatment [105].

Based on these data, and given the good tolerability of this repeated low-dose approach, metronomic chemotherapy may be considered a viable option also for metastatic TNBC, notably for low-burden TNBC patients and elderly patients.

## 7. Conclusions

This review demonstrates the complexity of the treatment for advanced TNBC, a very heterogeneous entity historically characterised by poor prognosis and a lack of effective therapeutic strategy. The possibility of affecting both the cancer cell and the microenvironment represents a great opportunity for the treatment of MTNBC, a tumour characterised by angiogenesis and an immunogenic microenvironment. The identification of new biomarkers, PDL1, AR, PI3K/AKT/mTOR and HER2-low expression, has led to the development of new therapeutic approaches, such as immunotherapy, targeted therapies and ADCs, in a constantly evolving scenario with a large amount of ongoing clinical trials on emerging therapies. Thus, to date, traditional chemotherapy—which still represents the most-used strategy to treat metastatic TNBC in clinical practice—can be placed side-by-side with new approaches, according to a precision medicine perspective. If immunotherapy or PARP inhibitors are already a consolidated reality, many of the drugs described in this review, such as sacituzumab govitecan and other ADCs, have just entered on the scene of TNBC treatment, and some therapies are still far from being used in clinical practice. Some drugs, such as larotrectinib and entrectinib for patients with TRK gene fusions and pembrolizumab for patients with MSI-high status or dMMR or high TMB, represent a treatment option also for metastatic TNBC, thanks to tumour agnostic approval, an approach based on cancer’s genetic and molecular features without regard to the tumour type. Although these molecular alterations are rare in TNBC, there exists a subgroup of patients that can benefit from these treatments. Among many new therapies, metronomic chemotherapy represents an opportunity for a subgroup of MTNBC patients, thanks to its tolerability and to its peculiar mechanism of action capable of affecting both the cancer cell and the microenvironment, notably for low-burden TNBC patients and elderly patients. Furthermore, a lot of clinical trials regarding well-known targets and new emerging biomarkers are still ongoing.

It seems to us from this extensive overview that, in the near future, TNBC patients should have many different options for treatment; however, the key questions are:What is, or will be, the optimal sequence with these agents? Most of these targets are not mutually exclusive, but each targeted therapy has been developed and tested as if its target was present alone. We strongly believe that the scientific community should make further effort to design strategic and multi-level decision trials, in order to allow all of us in the near future to adopt the most appropriate sequence;How will access to these drugs be regulated? In some cases—for example, PARP inhibitors—they could be the turning point for certain patients. This is why we hope that access will be as broad and rapid as possible, guaranteeing fair access and approval times in the various countries.

Although in the last few years the treatment of metastatic TNBC has made great strides, our current knowledge about biomarkers represents only the tip of the iceberg, and it is likely that, in the near future, the treatment landscape of TNBC will radically change again, with ever more precise perspectives of the medicine.

## Figures and Tables

**Figure 1 cancers-14-03729-f001:**
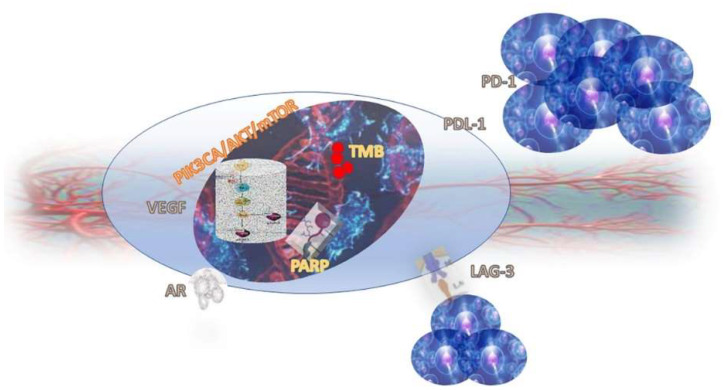
TNBC cell and its targets.

**Table 1 cancers-14-03729-t001:** Targeting the microenvironment (angiogenesis and the immune system): summary of clinical trials.

Target	Relevance in TNBC	Drugs	Clinical Trials (Phase)	Outcomes	Indication Approved/Not Approved
VEGFR	30–60%	bevacizumab	E2100 (phase III) (bevacizumab + paclitaxel vs. paclitaxel alone) [6]	PFS: 11.8 vs. 5.9 months in ITT population and 8.8 vs. 4.6 months in TNBC patients	Approved by EMA
			AVADO (phase III) (bevacizumab 7.5 mg/kg or 15 mg/kg + docetaxel vs. docetaxel alone) [7]	PFS: 9 vs. 10.1 vs. 8.2 months in ITT population	Not approved
			RIBBON-1 (phase III) (bevacizumab + CT vs. CT alone) [8]	PFS: 8.6 vs. 5.7 months for cape-based CT and 9.2 vs. 8 months for taxane-anthra-based CT	Not approved
			RIBBON-2 (phase III) (bevacizumab + CT vs. CT alone in II line) [9]	PFS: 7.2 vs. 5.1 months in ITT population, 6 vs. 2.7 months in TNBC patients	Not approved
			TANIA (phase III) (bevacizumab + CT vs. CT alone in II line) [10]	PFS2: 6.3 vs. 4.2 months	Not approved
			GINECO A-TaXel (phase II) (bevacizumab + paclitaxel + capecitabine single arm) [11]	ORR: 77%	Not approved
			Turandot (phase III) (bevacizumab + capecitabine vs. bevacizumab + paclitaxel) [12]	OS: 26.1 vs. 30.2 months in ITT population and 17.7 vs. 24.4 months in TNBC patients	Approved by EMA
			ATRACTIB (phase II) (bevacizumab + paclitaxel + atezolizumab single arm)	PFS	Ongoing
			NCT05192798 (phase II) (bevacizumab + nab-paclitaxel vs. nab-paclitaxel alone)	PFS	Ongoing
PD1/PDL-1	20%	atezolizumab	Impassion130 (phase III) (atezolizumab + nab-paclitaxel vs. nab-paclitaxel + placebo) [14]	PFS: 7.2 vs. 5.5 months in ITT population, 7.5 vs. 5.5 months in PD-L1 positive population OS: 25.4 vs. 17.9 months in PD-L1-positive population, not statistically significant in ITT population	Approved by EMA
			IMpassion131 (phase III) (atezolizumab + paclitaxel vs. paclitaxel + placebo) [15]	PFS: 6 vs. 5.7 months in PD-L1-positive population	Not approved
		pembrolizumab	Keynote-355 (phase III) (pembrolizumab + chemotherapy vs. placebo + chemotherapy) [16,17]	PFS: 9.7 vs. 5.6 months in CPS ≥ 1 population) OS: 23 vs. 16.1 months in CPS ≥ 1 population	Approved by FDA and EMA
			Keynote-119 (phase III) (pembrolizumab monotherapy vs. chemotherapy) [18]	OS: 12.7 vs. 11.6 in CPS ≥ 10 population, 10.7 vs. 10.2 in CPS ≥ 1 population, 9.9 vs. 10.8 in ITT population	Not approved
			Keynote086 (phase II) (pembrolizumab monotherapy in previously treated (cohort A) and untreated (cohort B) patients) [19]	Cohort A: ORR: 5.3% in ITT population, 5.7% in PD-L1-positive population OS: 9 months, PFS: 2 months, Cohort B: Safety: 63.1% AEs, no grade 4 events, ORR: 21.4%, PFS: 2.1 months; OS: 18 months	Not approved
		durvalumab + tremelimumab	NCT02536794 (phase II) (durvalumab + tremelimumab, single arm) [20]	ORR: 43% in TNBC	Not approved
		durvalumab + oleclumab	SYNERGY (NCT03616886) (phase Ib/II) (paclitaxel + carboplatin + durvalumab ± oleclumab)	AEs, CB	Ongoing
		leramilimab + spartalizumab	NCT02460224 (phase I/II) (leramilimab + spartalizumab, single arm) [21]	Dose limiting toxicity; durable responses in 2/5 TNBC patients	Not approved
TMB	<2%	pembrolizumab	keynote-158 (phase II) (pembrolizumab single arm) [22]	ORR: 34.3%	Approved by FDA
MSI-H/dMMR	<2%	pembrolizumab	keynote-158 (phase II) (pembrolizumab single arm) [22]	ORR: 34.3%	Approved by FDA

**Table 3 cancers-14-03729-t003:** ADCs in TNBC: summary of clinical trials.

Target	Relevance in TNBC	Drugs	Clinical Trials (Phase)	Outcomes	Indication Approved/Not Approved
HER2-LOW		trastuzumab deruxtecan	NCT02564900 (phase IB) (TDX-d in single arm) [80]	ORR: 44%, DoR: 10.4 months PFS: 11.1 months OS: 29.4 months	Not approved for TNBC
			Destiny-Breast04 (phase III) (TDX-d vs. CT) [81]	PFS: 9.9 vs. 5.1 months in ITT population and 8.5 vs. 2.9 months in TNBC patients	Not approved
			BEGONIA (phase IB-II) (TDX-d + durvalumab)	OS: 23.4 vs. 16.8 months in ITT population and 18.2 vs. 8.3 months in TNBC patients safety	Ongoing
TROP-2	78%	sacituzumab govitecan	IMMU-132-01 (phase I–II) (sacituzumab govitecan in single arm) [82]	ORR: 33.3%, DoR: 7.7 months	Approved by FDA and EMA
			ASCENT (phase III) (sacituzumab govitecan vs. CT) [83]	PFS: 5.6 vs. 1.7 months OS: 12.1 vs. 6.7 months	Approved by FDA and EMA
			SEASTAR (phase IB-II) (sacituzumab govitecan + rucaparib single arm)	Safety, ORR	Ongoing
			MORPHEUS-TNBC NCT03424005 (phase Ib/II) (sacituzumab govitecan + atezolizumab in single arm)	ORR, safety	Ongoing
			NCT04468061 (phase II) (sacituzumab govitecan + pembrolizumab vs. sacituzumab govitecan alone)	PFS	Ongoing
		datopotamab deruxtecan	NCT03401385 (phase I) (datopotamab deruxtecan in single arm) [84]	ORR: 34%	Not approved
			TROPION-Breast02 (phase III) (datopotamab deruxtecan vs. CT)	PFS, OS	Ongoing
			BEGONIA (datopotamab deruxtecan + durvalumab) (phase IB-II)	Safety	Ongoing
GPNMB	40%	glembatumumab vedotin	EMERGE (phase II) (glembatumumab vedotin vs. CT) [85]	ORR: 18% vs. 0% in ITT population and 40% vs. 0% in GPNMB over-expressing patients	Not approved
			METRIC (phase II) (glembatumumab vedotin vs. capecitabine) [86]	PFS: 2.9 vs. 2.8 months	Not approved
LIV-1	65%	ladiratuzumab vedotin	NCT01969643 (phase I) (ladiratuzumab vedotin in single arm)	ORR: 32% DCR: 64% CBR: 36%	Not approved
			NCT03310957 (phase IB-II) (ladiratuzumab vedotin + pembrolizumab)	ORR	Ongoing
HER3	17–43%	patritumab deruxtecan	NCT02980341 (phase I–II) (patritumab deruxtecan in single arm) [87]	ORR: 22.6%, DCR: 56.6%, DoR: 5.9 months	Not approved
			NCT04699630 (patritumab deruxtecan) (phase II)	ORR, 6 months PFS	Ongoing

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
