# Peer review of "Personalised Therapies for Metastatic Triple-Negative Breast Cancer: When Target Is Not Everything"

_cancers, 2022, doi:10.3390/cancers14153729_

Round 1

Reviewer 1 Report

This article highlights the targets and therapeutic options in triple negative breast cancer (TNBC). I enjoyed reading this manuscript. Language used to describe the science was elegant. The tables provided in this work citing different clinical trials for various biomarkers were informative and significant to the field.

I encourage the authors to include a figure demonstrating molecular mechanisms involved in TNBC pathogenesis and the biomarkers targeted by neoadjuvant therapies. That would be useful to the readers and elevate the quality of the work. And if possible, combine it with a pie chart differentiating the % of cancer subtypes in breast cancer and their respective treatment options (approved) in the diagram.

This work discussed the development of new biomarkers such as PDL1, AR, and HER2-low expression that gave birth to emerging therapeutic approaches such as immunotherapy, targeted therapies and ADCs. Specifically, I found the antibody drug conjugation (ADCs) molecule list interesting. Targeted protein degradation space (PROTACs) is currently evolving and I suggest the authors discuss a few lines about the molecule ARV-471, its associated clinical trial and the impact on ER+/HER2- breast cancer. Please speculate how do you see the protein degradation technology for TNBC?

My overall feedback is positive. I recommend this work for publication with minor changes. Congratulations to the authors for drafting this important work about TNBC. Best wishes.

Reviewer 2 Report

This review by Serena et al focuses on emerging personalized treatments for triple negative breast cancer (TNBC), some of which can be combined with traditional chemotherapies. The manuscript carefully reviews multiple targeting mechanisms, including angiogenesis, mechanisms regulating the immunogenic microenvironment, and key intracellular signaling pathways, and then ends with a description of metronomic chemotherapy. Each targeted system is effectively described, and the drugs that target the systems plus current clinical trials of the therapeutics are carefully reviewed. The text is set up logically with helpful tables that match the target with drug names and clinical trial information. Although there are numerous and recent reviews on therapeutics that target TNBC, this review does a good job combining descriptions of the therapeutics with current or pending clinical trials. Overall the manuscript is well written, but there are some sections with issues that should be addressed as follows.

Abstract, add "TNBC" to first mention of tripple negative breast cancer, and define ADCs.

Page 1, metastatic TNBC is shown as capital "M", but later written as small case, so this needs to be consistent.

Page 2, under 2.1, after the authors introduce bevacizumab, they need to simply define the general chemotherapy drugs mentioned throughout this section (and later), including paclitaxel, capecitabine, and docetaxel, to indicate how addition of bevacixumab acts in conjunction with these more commonly used chemotherapeutic agents. This description should also group the agents according to types, as later mentioned for taxanes and  anthrocyclines. Since many chemotherapeutics are described throughout the manuscript, perhaps a definition can be provided of the different chemotherapies in the Introductory comments on the first page, along with a table that summarizes the chemotherapeutic agents.

 Under 2.1, the authors need to define PFS and ORR (as done for OS in the Introduction) for the lay-reader.

Page 3, second paragraph and throughout the rest of the manuscript, define “pts” upon first use or just spell out "patients".

 Page 5, 3, first paragraph and sentence, "cytokines" is included with the list of cells in the tumor microenvironment but are not cells, so this needs rewording. Also, in the next sentence, define "TIL", and define pCR and DFS in the following sentence.

Page 7, third sentence, remove the ")" after TNBC (no parentheses needed).

Page 8, under 4.1, second paragraph, is "HR" intended to be "HRR"?

Page 10, paragraphs below section 4.3, several sentences are stand-alone and should be combined into the previous or subsequent paragraphs. Last sentence of the third paragraph is grammatically flawed and requires editing.

 Page 12, under 4.4, to be consistent with the other pathway descriptions, the authors should mention how NOTCH signaling is activated via cleavage of the transmembrane protein to release the Notch intracellular domain (NICD), which transports to the nuclease to act as a transcription factor.

There are multiple grammatical errors throughout the manuscript that must be fixed, so recommend a careful review for both spelling errors and grammatical mistakes. In addition, multiple sections include one-sentence paragraphs that should be edited to provide more cohesive paragraphs.

Reviewer 3 Report

The authors have written a comprehensive, well written review on personalised therapies for TNBC.

What is not clear to me is why in the title, they use the subphrase “When Target is not Everything”. In fact, in most of the review they focus on additional tragets for personalised TNBC therapy (‘emerging therapeutic targets and their related specific therapy’ as they call it themselves).

Concerning the content:

Could it be useful to mention some additional ongoing clinical trials/results in the context of TNBC that I found:

* concerning combination immunotherapies, Leramilimab (LAG525), an anti-LAG3 antibody, when combined with spartalizumab, an anti-PD-1 antibody, led to durable RECIST responses in a phase I trial, including 2 of 5 triple-negative breast-cancer (TNBC) patients (Hong DS, Schoffski P, Calvo A, et al. Phase I/II study of LAG525 ± spartalizumab (PDR001) in patients (pts) with advanced malignancies. J Clin Oncol. 2018;36(suppl 15):abstract 3012. doi: 10.1200/JCO.2018.36.15_suppl.3012.)

* The authors discuss PARP inhibitors. Could they also mention that clinical trials are ongoing evaluating for example glutaminase inhibitors such as CB-839 (eg NCT03057600)?

* NCT03719326 studies combination of chemotherapy with immunotherapies such as Adenosine Receptor Inhibitor Etrumadenant and PI3K Inhibitor IPI-549.

I have some minor grammar corrections/suggestions, such as:

* In “3. Targeting immune-system”, the hyphen is not required

* In “4. Targeting intracellular pathway”, I would suggest to change it to “4. Targeting intracellular pathways”, since multiple pathways are discussed.

* In  the conclusions, in “That’s why, we hope it will be as broad and rapid as possible, guaranteeing fair access and approval times in the various Countries”, a capital letter in not required for Countries.

Concerning the structure, the section on “2. Targeting angiogenesis” has only 1 subsection “2.1. VEGFR”. To avoid this, the authors could fuse section 2 and 3 into “Targeting microenvironment (angiogenesis and immune system)”, as they have also done for the Table 1.

Concerning formatting of the tables, the clarity could be improved by aligning the tekst to the top of each row. Now the text is lowered to the middle and as a consequence sometimes it is not fully clear where the new line starts among the various columns.
